# Astaxanthin for the Food Industry

**DOI:** 10.3390/molecules26092666

**Published:** 2021-05-02

**Authors:** Barbara Stachowiak, Piotr Szulc

**Affiliations:** 1Department of Technology of Plant Origin Food, Poznan University of Life Sciences, Ul. Wojska Polskiego 31, 60-624 Poznan, Poland; 2Department of Agronomy, Poznan University of Life Sciences, Ul. Dojazd 11, 60-632 Poznan, Poland; piotr.szulc@up.poznan.pl

**Keywords:** astaxanthin, carotenoids, xanthophylls, antioxidants, bioactive compounds, *Haematococcus pluvialis*, *Xanthophyllomyces dendrorhous*, crustacean byproducts, encapsulation

## Abstract

Xanthophyll astaxanthin, which is commonly used in aquaculture, is one of the most expensive and important industrial pigments. It is responsible for the pink and red color of salmonid meat and shrimp. Due to having the strongest anti-oxidative properties among carotenoids and other health benefits, natural astaxanthin is used in nutraceuticals and cosmetics, and in some countries, occasionally, to fortify foods and beverages. Its use in food technology is limited due to the unknown effects of long-term consumption of synthetic astaxanthin on human health as well as few sources and the high cost of natural astaxanthin. The article characterizes the structure, health-promoting properties, commercial sources and industrial use of astaxanthin. It presents the possibilities and limitations of the use of astaxanthin in food technology, considering its costs and food safety. It also presents the possibilities of stabilizing astaxanthin and improving its bioavailability by means of micro- and nanoencapsulation.

## 1. Introduction

Astaxanthin (3,3′-dihydroxy-β-carotene-4,4′-dione) is a carotenoid pigment without provitamin A activity in humans [1]. It is a xanthophyll with the chemical formula C_40_H_52_O_4_, molecular mass of 596.85 Da and density of 1.081 g/L. Its Chemical Abstracts Service (CAS) number is 472-61-7. Astaxanthin was first isolated from lobsters in 1938 [2].

Astaxanthin is commercially used, mostly in the feed industry. At present, together with canthaxanthin, it is the most important and the most expensive pigment used in aquaculture for pigmentation of salmon, trout and shrimp meat (these animals do not synthesize astaxanthin de novo), which affects consumers’ preferences around the world. Astaxanthin is a necessary component of the feed for aquarium fish as well as large ornamental fish. Scientific research has shown that the pigment has a positive influence on the color of egg yolk as well as the skin and meat tissue of broiler chicken carcasses. Due to strong anti-oxidative properties and other health benefits, astaxanthin is also used in the nutraceutical and cosmetic industries, and in some countries, it is occasionally used to fortify foods and beverages [3,4,5,6].

Chemical synthesis of astaxanthin is currently the most cost-effective, and thus its synthetic preparations have dominated over 95% of the feed market [7]. The pharmaceutical, cosmetic and food industries use only natural astaxanthin. According to Grand View Research, the global astaxanthin market size was estimated at USD 1.0 billion in 2019. It is expected to witness a compound annual growth rate of 16.2% from 2019 to 2027 to reach USD 3398.8 million by 2027, owing to rising awareness of natural astaxanthin and its well-documented, multifunctional health benefits and safety [8].

This fact, as well as the coloring properties of the pigment, enable food technologists to design a sensorily attractive assortment of functional foods and active packages. There are two reasons why the use of astaxanthin in food technology is limited. As natural and synthetic astaxanthin differ in their structure, it is not certain how long-term consumption of synthetic astaxanthin may affect human health. The other reason is the few sources of natural astaxanthin and thus the high cost of obtaining it.

This review presents the possibilities and limitations to the use of astaxanthin in food technology. The article characterizes the structure, health-promoting properties and industrial use of astaxanthin. It gives an overview of industrial and potential sources of the pigment and considers costs and food safety. It also presents the possibilities of stabilizing astaxanthin and improving its bioavailability by means of micro- and nanoencapsulation.

## 2. The Occurrence, Structure and Industrial Potential of Astaxanthin

In nature, astaxanthin can be found in aquatic environments. It gives pink and red colors to the meat of fish such as Atlantic salmon, rainbow trout, Arctic char and red bream and to the shells of crustaceans such as krill, shrimp and lobster, etc. as well as the feathers of some wading birds, e.g., flamingo, scarlet ibis [9,10]. In the natural environment, the color of these animals is the result of bioconcentration of the pigment at consecutive trophic levels in the food chain [11]. In the aquatic environment, astaxanthin can be found in algae, which can synthesize this pigment, as well as in plankton crustaceans, which are capable of astaxanthin conversion from carotenoid precursors (chiefly from β-carotene and zeaxanthin). Thus, the color intensity of animal tissues mostly depends on the presence of astaxanthin in these animals’ diets. This fact significantly influences the use of this pigment in the feed industry.

Like most carotenoids, astaxanthin is a 40-carbon tetraterpene consisting of linked isoprene units. The molecular structure of astaxanthin is composed of a linear polyene chain and two terminal β rings (Figure 1) [12]. The system of 11 conjugated double bonds determines the pink and red color of astaxanthin (absorption maximum: in dimethyl sulfoxide (DMSO)—492 nm, in acetone—477 nm, in methanol—477 nm, in dimethylformamide—486 nm, in chloroform—86 nm) and is responsible for its anti-oxidative potential [13]. Apart from that, both terminal rings of astaxanthin contain two polar function groups: hydroxyl (OH) located at the two asymmetric carbons C3 and C3′ and keto (=O) at carbons C4 (Figure 1).

The presence of these groups is typical of astaxanthin and makes it unique among other carotenoids. Thanks to the polar–non-polar structure, astaxanthin can fit the hydrophobic polyene carbon chain inside the bilayer lipid cell membrane, and its polar terminal rings can be located near its surface (Figure 2). In consequence, in comparison with other carotenoids, astaxanthin exhibits very high anti-oxidative activity in lipid systems [14].

Astaxanthin was found to protect membrane phospholipids and other lipids from peroxidation more effectively than β-carotene and lutein [16,17]. Its anti-oxidative activity is 10 and 100 times higher than that of β-carotene and vitamin E, respectively [18,19]. The outstanding anti-oxidative potential of astaxanthin has encouraged numerous investigations which indicated its potential clinical use in the prevention and treatment of diseases associated with reactive oxygen species such as cancers [20,21], neurodegenerative diseases [22,23,24], eye diseases (cataract, macular degeneration, asthenopia) [25,26], atherosclerosis and type 2 diabetes [15,27,28,29]. Astaxanthin counteracts gastric inflammations caused by *Helicobacter pylori* (chronic type B gastritis, peptic ulcer disease and gastric carcinoma) [30] and inflammations of the vocal folds [31]. It might be used to treat clinical sepsis [32]. It exhibits an immunomodulatory effect [33,34]. In contrast to β-carotene, astaxanthin easily permeates the blood–brain barrier as well as the blood–retinal barrier and prevents inflammations of these organs [12]. Astaxanthin may prevent photooxidative processes caused by UV radiation [17]. It improves the condition of men’s and women’s skin when administered orally. It reduces the depth of wrinkles, reduces age spot size and improves elasticity, skin texture, moisture content in the corneocyte layer and the corneocyte condition [35]. It is a bioactive component of cosmetics (creams, balms, oils, anti-aging serums), providing protection from solar radiation. Natural astaxanthin producers recommend a daily dose of 4–12 mg for health benefits, which is similar to other carotenoids. The immunomodulatory effect was achieved in clinical trials when the daily dose of astaxanthin was 2 mg [33]. The EFSA Panel on Dietetic Products, Nutrition and Allergies recommends that the maximum daily dose of astaxanthin from alga *Haematococcus pluvialis* (AstaREAL supplements) should not exceed 4 mg (0.06 mg/kg bw per day for a 70-kg person) [36]. Research has not shown that it is possible to overdose on astaxanthin. For example, Buesen et al. [37] did not observe any negative effects of the pigment when it was applied to rats at doses of 700–920 mg/kg/bw. Contrary to other antioxidants, astaxanthin never becomes a pro-oxidant [38].

Depending on the configuration of hydroxyl groups at the asymmetric carbons C3, different configurational isomers of astaxanthin can be formed: the (3R, 3′R) and (3S, 3′S), which are enantiomers, and the meso form (3R, 3′S) (Figure 1). Astaxanthin diastereoisomers differ in their physicochemical and biological properties as well as bioavailability. All of the aforementioned astaxanthin stereoisomers can be commonly found in nature. Their share depends on the source where they are found. The accumulation of astaxanthin isomers in aquatic animals is related to the isomer configuration of dietary astaxanthin (Table 1).

The presence of hydroxyl groups in benzoid rings enables esterification of astaxanthin. Esterified astaxanthin is more resistant to temperature fluctuations and photochemical reactions (photolysis, photosensitized oxidation) than free astaxanthin. In nature, astaxanthin can usually be found in the form of mono- and diesters, e.g., in algae and crustaceans’ shells [39]. According to Snoeijs and Häubner [40], in natural zooplankton communities in the Baltic Sea, diesters prevailed during the cold season, but monoesters prevailed in the warm season. Astaxanthin can naturally be found in complexes with proteins or fats. The shells of lobsters, shrimps and other crustaceans contain a bright blue astaxanthin complex with a protein, i.e., crustacyanin. Only after thermal processing (after protein denaturation) is astaxanthin released, and the typical pink color can be seen. The dark green color is astaxanthin lipoglycoprotein, present in lobster ovaries and eggs. Ovorubin is a complex of astaxanthin ester with glycoprotein that gives the red color to the eggs of channeled apple snails (*Pomacea canaliculata*) [41].

Long thermal processing, even at low temperature, promotes the hydrolysis of esterified astaxanthin and yields free astaxanthin. This fact is very important during the production of smoked salmonoids and dried salted shrimp. Studies showed that the astaxanthin content in cooked shrimp decreased by 78% after four days of direct sun drying due to photodegradation [46]. The content of this pigment was much lower than the content obtained in a jet-spouted bed-drier at 80, 100 and 120 °C [47].

Apart from pigmentation of animal organisms, like other carotenoids, astaxanthin has other metabolic and physiological functions. It has a positive influence on the growth and reproduction of crustaceans [48], sea urchins (*Pseudocentrotus depressus*) [49], guppies (*Poecilia reticulata*) [50] and salmonids. A positive effect of astaxanthin supplementation on the reproductive traits of rainbow trout was found. In this case, the astaxanthin content in the eggs and the fertilization rate, the percentage of eyed eggs and hatching were significantly correlated [51]. Studies have shown that astaxanthin supplementation contributes to the health of laying hens by influencing the activity of antioxidant enzymes as well as anti-inflammatory and immunomodulating interleukins. Astaxanthin improves superoxide dismutase (SOD) and glutathione peroxidase (GSH-Px) activities and diminishes malondialdehyde (MDA) content in both the liver and serum. Additionally, astaxanthin alleviates interleukin 2, 4, and 6 (IL-2, IL-4 and IL-6, respectively) in serum [52].

## 3. Commercial Sources of Astaxanthin

### 3.1. Chemical Synthesis

There are a lot of advantages of the chemical synthesis of carotenoid pigments, e.g., it is possible to obtain pigments of desired purity and consistency. However, as a consequence, mixtures of stereoisomers are produced. Some of them cannot be found in nature. Therefore, they may not exhibit activity identical to natural carotenoid isomers. They may exhibit different activities from natural carotenoids and cause local adverse effects [53]. It is noteworthy that astaxanthin synthesized in nature occurs in the trans form (3S, 3S), whereas synthetic astaxanthin is a mixture of two optical isomers and the meso form at a ratio of 1:2:1 (3R, 3′R), (3R, 3′S) and (3S, 3′S) (Figure 1) [54]. As farm animals are fed with synthetic astaxanthin, it is easy to determine their origin by analyzing the composition of the pigment.

Until now, chemical synthesis has been the cheapest method of obtaining astaxanthin because it does not consume much energy and emits small amounts of greenhouse gases. The cost of production of 1 kg of synthetic astaxanthin is estimated at about 1000 dollars, whereas its market value is higher than 2000 dollars. 1 kg of the natural pigment obtained from *Xanthophyllomyces dendrorhous* yeasts costs 2500 dollars, whereas 1 kg of astaxanthin from *Haematococcus pluvialis* algae costs 7000 dollars [55]. Natural astaxanthin is characterized by higher oxygen radical absorbance capacity, higher stability and better assimilability. This difference results from the presence of astaxanthin stereoisomers in synthetic preparations [56,57]. For this reason, at present, only natural astaxanthin can be a component of dietary supplements for humans.

The oldest strategy for chemical synthesis of astaxanthin involves the Wittig reaction of two appropriate C15-phosphonium salts with a symmetrical C10-dialdehyde as a central building block [58]. Nguyen [59] also lists other strategies: cantaxanthin hydroxylation, C10+C10+C10 synthesis via dienol ether condensation and isomeration of lutein extracted from marigold to astaxanthin.

In the early 1980s, Hoffmann La Roche (Basel, Switzerland) developed and manufactured the first synthetic astaxanthin preparation under the trade name CAROPHYLL^®^Pink (C^®^P). In C^®^P, sensitive astaxanthin molecules are stabilized with antioxidants and embedded in a carbohydrate and gelatin matrix. This stable product is coated with starch to improve handling. According to the manufacturer’s declaration, C^®^P contains 10% astaxanthin. In 2002, Hoffmann La Roche sold the production rights to DSM (Denmark), who developed CAROPHYLL^®^ Stay-Pink (C^®^SP), containing about 11% astaxanthin dimethyl succinate. Orally administered, it is hydrolyzed and converted to free astaxanthin in the intestines of fish and then absorbed, metabolized and distributed in the same manner as free astaxanthin. The purpose of C^®^SP is to provide farmed Atlantic salmon (*Salmo salar*) and rainbow trout (*Oncorhynchus mykiss*) with a source of the carotenoid astaxanthin, which gives the characteristic pink color to wild salmonids.

Lucantin^®^ Pink (BASF Chemical Company, Ludwigshafen, Rhineland-Palatinate, Germany) is another synthetic astaxanthin preparation that contains at least 10% astaxanthin. It is used for efficient pigmentation of shrimp, salmon, egg yolks and broiler skins.

In May 1995, the Food and Drug Administration (FDA) permitted the use of synthetic astaxanthin as a pigment in animal and fish feeds [60]. In the European Union, synthetic astaxanthin was registered as 2a E161j. The use of synthetic astaxanthin for feeding salmon, trout, shellfish and ornamental fish is permitted without expiry date. The dose of the pigment cannot be higher than 100 mg/kg of the total mixed ration (moisture content 12%) [61].

Apart from that, the EU Commission Regulation (EC) No. 393/2008 permits the use of astaxanthin dimethyldisuccinate (2a(ii) 165) as a feed additive for salmon and trout [62]. In July 2020, the European Commission renewed the authorization of astaxanthin dimethyldisuccinate as a feed additive for fish and crustaceans and repealed Regulation (EC) No. 393/2008 [63].

The EU Regulation (EC) No. 1925/2006 on the addition of vitamins, minerals and other substances to foods does not permit the use of synthetic astaxanthin in food [64]. Nor does it have GRAS status (Generally Recognized as Safe) in the US.

These legal restrictions were introduced because of the differences between natural and synthetic astaxanthin, as the latter might be harmful to humans and the environment. A cause and effect relationship has not been established between the consumption of astaxanthin and the maintenance of normal joints, tendons or connective tissue; protection of DNA proteins or lipids from oxidative damage; maintenance of normal visual acuity; and maintenance of normal blood cholesterol concentrations or maintenance of low plasma concentrations of CRP (C Reactive Protein) [65].

### 3.2. Natural Systems as a Source of Astaxanthin

Astaxanthin is obtained from primary sources such as higher plants; microscopic phytoplankton algae *Haematococcus pluvialis* [2], *Chlorella zofingiensis, Chlorococcum* sp. [66]; and some microorganisms, i.e., *Xanthophyllomyces dendrorhous* (anamorph *Phaffia rhodozyma*) yeasts and bacteria such as *Mycobacterium lacticola, Brevibacterium*, *Agrobacterium aurantiacum, Alcaligens sp.* strain PC-1. and *Paracoccus carotinifaciens* [10,67]. The industrial production of natural astaxanthin started in the 1980s. Cynotech Corporation (Kona, HI, USA) is the oldest and largest producer of the pigment from microalgae. The trade name of the product is BioAstin^®^. It is oleoresin extracted from *Haematococcus pluvialis*. It includes a minimum of 4 milligrams of astaxanthin per gel cap. In China, astaxanthin in industrial production is extracted from krill and crustacean byproducts.

#### 3.2.1. Astaxanthin from Plant Systems

Species of the *Adonis* genus, i.e., *A. aestivalis* and *A. annua* (Figure 3a), are the richest sources of astaxanthin. According to Cunningham and Gantt [68], this pigment makes up about 1% of the dry matter of petals of these plants.

However, due to the low yield of flower biomass from the cultivation area, this plant is not a cost-effective source of the pigment. Early studies on using *Adonis* species as industrial sources of astaxanthin concerned the obtaining of cultivars with a larger number of petals in the flower head [70]. Later studies concerned the isolation of genes encoding the astaxanthin biosynthesis pathway from *Adonis* plants and their transfer to other plants, e.g., marigolds, which guaranteed a high yield of biomass with carotenoids [71]. As the synthesis of carotenoids depends on the content of precursors and the possibility of converting them in the conversion pathway, researchers became particularly interested in plants that can produce large amounts of β-carotene, such as marigolds, various oil palm species, canola rapeseed, sweet potatoes and maize because it is possible to insert DNA fragments responsible for the conversion of β-carotene into astaxanthin from *Adonis* plants into the genome of these plants [72,73]. Typically, the biosynthesis of astaxanthin from β-carotene requires ketolase and hydroxylase to add carbonyl and hydroxyl at positions 4 and 3 of each terminal β-ionone ring, respectively. Researchers successfully produced transgenic tomatoes with a high concentration of free astaxanthin in leaves (3.12 mg/g) and its esterified form in fruit (16.1 mg/g). The success was achieved through co-expression of two genes from microalgae, i.e., β-carotene ketolase from *Chlamydomonas reinhardtii* and β-carotene hydroxylase from *Haematoccocus pluvialis* [74]. There are also studies on genetic modification of marigolds (*Tagetes*) and their use as a source of astaxanthin for feeding animals [75].

#### 3.2.2. Microbiological Synthesis of Astaxanthin

At present, microbiological synthesis of astaxanthin is one of the most intensely developing research areas. It has more advantages than plant production. It is easy to culture microorganisms, which grow fast on cheap culture media. Their development does not depend on weather conditions, and the shade of the pigment is stable [76]. However, the use of microbial systems for astaxanthin production is not very economical. The pigment is an intracellular metabolite. Therefore, the cost of biosynthesis depends on the cost of biomass production, the concentration of the pigment in cells, the metabolic activity of the cells producing the pigment and the need to isolate the pigment from the cells and purify it [53].

*Haematoccocus pluvialis* freshwater microalgae are a basic source of natural astaxanthin on the market. They accumulate up to 4% of the pigment in dry biomass. It is the highest natural concentration of astaxanthin. The predominant form of astaxanthin in *H. pluvialis* is monoester [2]. Researchers stress the fact that it is particularly difficult to breed microalgae in open systems, e.g., in ponds, due to the risk of contamination with unwanted species of algae, bacteria, fungi, etc. For this reason, it is necessary to use expensive, high-capacity photobioreactors [77]. Another problem is that *Haematoccocus pluvialis* grow slowly, and their yield of biomass is low when they are grown on traditional media. Apart from that, they accumulate astaxanthin only when they are exposed to environmental stress, i.e., a nitrogen or phosphorus deficit, the presence of salicylic acid and ethanol, high salinity of the growth medium or intensive light [2,78,79]. In such cases, they form immotile thick-walled hematocysts that contain the pigment (Figure 3b). Thirdly, in order to isolate astaxanthin from hematocysts, it is necessary to disintegrate the thick cell wall. All these procedures make astaxanthin very expensive. Its production is limited to specialized markets, chiefly the pharmaceutical market. Astaxanthin is produced from *Haematoccocus pluvialis* microalgae in the United States, Japan and India [80]. It is a component of vitamin preparations, dietary supplements and protective creams. It is also used in organic farming. In 2014, the EFSA Panel on Dietetic Products, Nutrition and Allergies of the European Commission issued a positive opinion about the safety of astaxanthin-rich ingredients of AstaREAL preparations made from *Haematoccocus pluvialis*. According to the opinion, their consumption was not considered to be nutritionally disadvantageous and there were no safety concerns regarding genotoxicity [36]. In December 2017, astaxanthin-rich oleoresin from *Haematococcus pluvialis* algae was on the European Commission list of novel foods [81]. *H. pluvialis* cultivation can be carried out both in closed systems exposed to the sunlight or strictly controlled lighting, or in open ponds. Supercritical fluid extraction (SFE) is used to remove astaxanthin oleoresin from dried algal cells with supercritical CO_2_ or ethyl acetate as solvents The technique for SFE is described below. The astaxanthin is diluted in edible oils such as olive oil, safflower oil, sunflower oil or MCT (Medium Chain Triglycerides) at six levels: 2.5%, 5.0%, 7.0%, 10.0%, 15.0% and 20.0%. These preparations are intended to be used in fermented and non-fermented liquid dairy products, fermented soya products and fruit drinks for healthy adults [36].

For many years, scientific researchers and industrial producers have been particularly interested in red *Xanthophyllomyces dendrorhous* yeasts (Figure 3c), which synthesize unesterified astaxanthin, predominantly in the (3R, 3R′) form, which is different from that of *Haematoccocus pluvialis* algae. It is concentrated with other carotenoids in lipid droplets suspended in the cytoplasm or in cytoplasmic membranes in the lipid layer (it is not visible in microscopic studies). This form is very stable [82].

Studies indicated that unesterified astaxanthin was more efficiently taken up and utilized for pigmentation in rainbow trout than astaxanthin dipalmitates, probably due to the limited capacity of intestinal esterases to hydrolyze these esters [83]. It is particularly important because inactivated yeast biomass can be a ready product, which is not only rich in assimilable astaxanthin, but also necessary nutrients—proteins, lipids and B vitamins [11].

Low cellular concentration of astaxanthin is an essential problem while using *Xanthophyllomyces dendrorhous* yeasts for industrial production of the pigment. The content of astaxanthin in wild strains ranges from 0.01 to 0.03% of dry matter. As the cellular concentration of the pigment is so low, it is necessary to add a few percent of yeast to feeds so as to give the right color to salmon or trout meat. However, it is not advisable in aquaculture due to the high content of polysaccharides in the yeast cell walls [84]. Only the *Xanthophyllomyces dendrorhous* strains capable of astaxanthin synthesis 5- to 10-fold higher than in wild strains may be used for industrial production of the pigment and greatly reduce its market price [82,85]. Such strains can be obtained with chemical mutagenesis [86] or with a combination of classical mutagenesis and genetic pathway engineering [87,88].

#### 3.2.3. Crustacean Byproducts

Inedible parts of shrimp, crabs and other crustaceans, i.e., heads, shells, tails, etc., can be used as sources of natural astaxanthin in aquaculture. According to Mezzomo et al. [89], the annual worldwide capture of marine crustaceans was 3.2 million tons. Inedible byproducts made up 40–56% of the raw material weight, depending on the species, size and shelling procedure [90,91]. Non-polar solvents and vegetable oil are routinely used for industrial extraction of carotenoids, including astaxanthin, from crustacean byproducts. The choice of solvent is very important because it affects the astaxanthin extract quality. The solvent must be non-flammable, non-toxic, non-volatile and effective at low temperatures. Sunflower, groundnut, coconut and rice bran oils seem to be the most adequate. In recent years, flaxseed oil has gained considerable attention as a potential astaxanthin extractant due to the high content of omega-3 acids, i.e., alpha-linolenic acid and linoleic acid [92]. These acids have been shown to provide protection against cardiovascular disease and inflammation. Dispersed natural astaxanthin in flaxseed oil may provide healthier food options for consumers. Pu et al. [91] indicated that astaxanthin extracted from shrimp byproducts by means of flaxseed oil effectively protected the fatty acids it contained from oxidation during heating from 40 to 60 °C for 4 h. Nevertheless, the rate of astaxanthin degradation in flaxseed oil was significantly influenced by temperature. Astaxanthin was stable in flaxseed oil only at 30 and 40 °C but exhibited substantial degradation at 50 and 60 °C. In the near future, astaxanthin preparations in flaxseed oil may become as popular as krill oil (e.g., Gold Krill, Mega Red Omega 3, Krill Oil), which is a source of astaxanthin and omega-3 acids. These preparations are recommended to prevent atherosclerosis, cardiovascular, eye and neurodegenerative diseases caused by aging [91].

Higher efficiency of astaxanthin extraction from crustacean byproducts can be achieved by using organic solvents such as chlorinated ones. However, they are toxic and potentially carcinogenic. On the other hand, solvents used for industrial production such as *n*-hexane, *n*-heptane, acetone, methanol and petroleum ether require high temperatures, affecting thermolabile astaxanthin. Otherwise, the pigment extraction efficiency is low [90,93].

Supercritical fluid extraction (SFE) can be used as an alternative to conventional techniques of carotenoid extraction from crustacean byproducts. The use of nonpolar supercritical carbon dioxide (SC-CO_2_) as the solvent is the most commonly employed in SFE for extracting low-polarity and heat-sensitive bioactive compounds due to its low critical properties (Tc = 31.1 °C; Pc = 73.8 bar) [94]. SC-CO_2_ is chemically inactive, available, economical and non-toxic. It is not necessary to remove the solvent when CO_2_ is used in a supercritical state because this gas is at a normal temperature and atmospheric pressure. Apart from that, CO_2_ has GRAS status, and it is not expensive [89,90]. Currently, SC-CO_2_ is the most modern and effective method of obtaining essential oils, polyunsaturated oils, phytosterols, carotenoids, flavonoids and anthocyanins [95]. Sánchez-Camargo et al. [90] extracted astaxanthin from freeze-dried red spotted shrimp (*Farfantepenaeus paulensis*) waste (including the head, tail and shell) using SC-CO_2_ and evaluated the effects of the extraction conditions of pressure (200–400 bar) and temperature (40–60 °C) on the total extraction yield, astaxanthin extraction yield and astaxanthin concentration in the extract. It was shown that temperature, and especially pressure, had a significant effect on the astaxanthin extraction yield. The highest amount of the extract (with 39% astaxanthin recovery) was obtained at 43 °C and 370 bar. At low pressures, an increase in temperature resulted in a decrease in the amount of astaxanthin extracted.

Mezzomo et al. [89] studied SFE efficiency in order to concentrate carotenoids from pink shrimp (*Penaeus brasiliensis* and *Penaeus paulensis*) processing waste (composed essentially of head, carapace). The process efficiency was studied by the effects of the operational conditions and co-solvents, hexane:isopropanol solution (50:50, *v*/*v*) and sunflower oil (as co-solvents mixed to the supercritical CO_2_ in concentrations of 2 and 5% (*w*/*w*)). The highest astaxanthin yield was obtained with SC-CO_2_ at 300 bar and 60 °C. Although the use of hexane:isopropanol solution in SFE was successful to enhance the extraction yield compared to SFE with SC-CO_2_, the system selectivity did not increase carotenoid concentration. The authors of both studies indicated that carotenoid extraction increased along with CO_2_ density. The pigment extraction efficiency was lower at lower pressures and higher temperatures.

However, there are some limitations to the use of crustacean byproducts as a basic source of astaxanthin for aquaculture. Apart from the seasonal availability of crustacean byproducts (e.g., in Asian countries), the high costs of their storage and the need to protect them from decay (usually by mild lactic acid fermentation or with organic acids), these byproducts have a low content of astaxanthin—about 0.15%. Therefore, their content in feeds must be high (10–25%) to achieve adequate color of animal tissues. Unfortunately, they contain large amounts of water, ash and chitin, which limit their quantitative share in feed [2]. Additionally, even low concentrations of organic acids used for preservation cause the conversion of astaxanthin monoesters into diesters and reduce the amount of carotenoids recovered from byproducts. However, Sachindra et al. [96] indicated that lactic acid fermentation reduced the amount of solvent (organic compound or vegetable oil) used for the isolation of carotenoids from shrimp byproducts.

## 4. Methods of Astaxanthin Stabilization and Improvement of Its Bioavailability

Humans usually consume synthetic astaxanthin in seafood including farmed salmonids and shrimp. Food producers are particularly interested in astaxanthin due to its anti-oxidative potential and attractive color. The supplementation of astaxanthin in food might be a practical and beneficial health management strategy. The following products are mostly taken into consideration: fermented and non-fermented liquid dairy products, fermented soya products and fruit drinks for healthy adults at a maximum level of 1.6 mg astaxanthin per 100 g or 100 mL [36].

Apart from the price, there are two reasons why the use of natural trans astaxanthin in the food industry is limited. Firstly, it is unstable during isolation, manufacture and storage. Due to the highly unsaturated structure of astaxanthin, it can be easily damaged under adverse conditions of the technological process, e.g., acidic environments, heat, light, transition metal ions, singlet oxygen and free radicals, especially after being removed from its biological matrix [36]. This can cause the loss of its desirable nutritive and biological properties as well as the production of undesirable flavor and aroma compounds. The other serious problem which limits industrial use of astaxanthin is its poor solubility in water (83 mg/L) as well as its limited solubility in lipid blood components, e.g., triglycerides. As a consequence, astaxanthin is characterized by very low bioavailability, similar to other functional lipophilic nutrients [97]. As the solubility of bioactive compounds determines their bioavailability, the slow dissolution or solubilization of functional lipophilic nutrients in aqueous-based systems results in their low absorption rate and their low bioavailability [98]. For this reason, there has been research on the production of hydrophilic stable and bioactive derivatives of the pigment, e.g., various esters of astaxanthin such as disodium disuccinate astaxanthin, tetrasodium diphosphate astaxanthin and various fatty acid esters of astaxanthin [99,100,101].

The introduction of astaxanthin and other carotenoids into aqueous food systems and other complex environments usually causes the formation of oil-in-water (O/W) emulsion. Another step involves protecting the dispersion of astaxanthin from molecular modification and destruction. Many studies indicate that high stability, water solubility and high bioavailability of astaxanthin dispersion can be easily achieved by encapsulation technologies—microencapsulation and nanoencapsulation systems. In order to achieve the encapsulation of bioactive food compounds, polymer carriers are usually used. They should be compatible with the product’s properties (flavor, texture, shelf life), biodegradable and easy to use.

Proteins, especially milk proteins, are good emulsifiers, and hence, they are utilized as ingredients in a wide range of formulated food emulsions. Anarjan et al. [102] used sodium caseinate to stabilize astaxanthin nanodispersion. Dispersions were prepared using an emulsification–evaporation technique. The researchers obtained a preparation with optimal physicochemical properties (average particle size, polydispersity index, minimal loss of astaxanthin during the process). Astaxanthin nanoemulsion was prepared by three passes through a high-pressure homogenizer at 30 MPa. Next, the organic solvent (dichloromethane) was removed from the system by evaporation at 25 °C.

Shen and Quek [103] suggested using spray-drying technology to produce high-quality encapsulated astaxanthin that could be used in food systems. Encapsulation with spray-drying technology consists of homogenization of lipophilic core materials in a solution containing wall material to form a stable emulsion. Next, the emulsion is fed into a spray dryer, where it is converted into dry powders. Researchers have also used natural emulsifiers accepted by consumers, whey protein isolate and sodium caseinate, with soluble corn fiber as wall systems. The core material contained sunflower oil and the commercial astaxanthin preparation Cyanotech Bioastin. After homogenization and spray- drying of the emulsion, the powdered, encapsulated astaxanthin preparation was characterized by acceptable properties including water activity, surface morphology and oxidative stability. The microencapsulation efficiency was high (~95%) for both types of wall systems, indicating the suitability of these hydrophilic wall matrices for the encapsulation of hydrophobic astaxanthin.

Polysaccharides and their derivatives have also received great acceptance from the pharmaceutical and food industries as emulsion stabilizers due to their safety, biodegradability, biocompatibility and non-toxicity. Higuera-Ciapara et al. [104] obtained a thermostable astaxanthin preparation (within a temperature range of 25–45 °C during eight-week storage) that was characterized by good solubility. They applied microencapsulation of the pigment emulsion in a chitosan matrix cross-linked with glutaraldehyde, using the method of multiple emulsion/solvent evaporation. Cyclodextrins (CDs) are cyclic oligosaccharides (oligodextrins) obtained by the enzymatic breakdown of starch. They consist of various numbers of glucopyranose residues, linked together by α-1,4-glycosidic bonds in order to form a toroidal ring shape, with well-established use as encapsulating agents. The inner surface of the taurus is hydrophobic. The outer surface is hydrophilic, which makes the CDs well-soluble in water. Due to their molecular structure, CDs form stable inclusion complexes (host–guest complexes) with many molecules and organic compounds. The best-known CDs are α-CD, β-CD and γ-CD, which are composed of 6, 7 and 8 glucopyranose units, respectively [94]. CDs differ in internal diameter and are therefore characterized by selectivity of complexation. Among them, the most important in terms of practical use is β-CD, and its derivatives grade improve the water solubility of lipophilic small molecules, including carotenoids, and are widely used to prepare encapsulated substances in food and pharmaceutical applications [105]. Chen et al. [97] showed that in comparison with native astaxanthin, the stability of the inclusion complex of astaxanthin with β-CD (1:4) against temperature and light was greatly enhanced, but its aqueous solubility was only slightly enhanced (<0.5 mg/mL). Yuan et al. [106] obtained an inclusion complex of astaxanthin with hydroxypropyl-β-cyclodextrin (HP-β-CD) that was characterized by high astaxanthin solubility (>1.0 mg/mL). HP-β-CD protected astaxanthin from thermal degradation up to 40 °C. The complexation of crystalline astaxanthin with a derivatized form of β-CD and a solubilizing agent, Captisol^®^ (sulfobutyl ether β-cyclodextrin (sodium)), increased the water solubility of astaxanthin by approximately 71-fold to a concentration of 2 μg/mL [101]. According to Lancrajan et al. [107], carotenoid incorporation into a β-cyclodextrin carrier was more efficient than liposomal delivery or traditional methods of dissolving the carotenoid in an organic solvent such as tetrahydrofuran (THF), ethanol, dichloromethane or chloroform. The potential breakdown of cyclodextrin to individual sugar moieties is less toxic to humans than organic solvents.

At present, the largest number of studies concern the incorporation of astaxanthin in a nanodispersion system. Nanodispersion systems seem to be useful in many food and pharmaceutical applications because of their high stability, water solubility and high bioavailability as well as their ease of processing [108]. There is no clear definition of the term ‘nano’ as applied to foods. By reducing the particle size below a certain threshold value, the resulting material exhibits physical and chemical properties (color, solubility, chemical reactivity and toxicity) that are significantly different from those observed in their macroscopic counterparts. Thus, the scale of particle size reduction determines its applicability. Therefore, it is difficult to define the upper range of the nanostructure size. Many scientific publications use the term ‘nano’ to describe structures of about several hundred nanometers (instead of strictly 100 nm) [109]. The European Commission recommends that the term nanomaterial should be used for a material containing particles in an unbound state or as an aggregate or agglomerate and where, for 50% or more of the particles, one or more external dimensions are within the size range of 1–100 nm [110]. Due to their small size and high surface–volume ratio, nanoparticles provide great potential for the food, pharmaceutical and cosmetic industries. However, there is concern about long-term risks related to hard nanomaterials because these insoluble and indecomposable nanoparticles may accumulate in target organs.

The emulsification–evaporation technique is usually applied to prepare stable nanodispersion of astaxanthin. At the first stage, the pigment is solved in a lipophilic organic solvent. The following astaxanthin solvents are routinely used in studies: dimethyl sulfoxide (DMSO), THF, acetone, methanol, ethanol, acetonitrile, dichloromethane and chloroform. They are toxic to humans and need to be removed from the final product. Apart from that, their presence causes the isomerization of all-[E] natural astaxanthin to [Z]-isomeric forms, especially to 9-[Z] and 13-[Z] isomers (Figure 4). In consequence, the color becomes lighter [47].

[Z]-isomers are characterized by lower bioavailability and stability, and they oxidize more rapidly than all-[E]-isomers when exposed to light, oxygen or high temperature. This is important in technological processes, e.g., the feed production process (extrusion) [42]. On the other hand, they exhibit higher anti-oxidative activity than trans isomers, especially 9-[Z]-astaxanthin isomer [18]. Its DPPH scavenging activity, inhibition effect on lipid peroxidation and ROS generation in human neuroblastoma SH-SY5Y cells were higher than of all-trans astaxanthin. The study by Yuan and Chen [111] showed that the degree of isomerization of the pigment depended on the solvent—it was low for DMSO, but chlorinated solvents strongly promoted astaxanthin isomerization, and therefore, they should be avoided. Astaxanthin isomerization increased along with temperature (within the range of 25–50 °C). The authors indicated that due to food safety, vegetable oil was the best for the isolation of astaxanthin for nutritional purposes and that the isolation should take place at room temperature.

At the next stage, an oil-in-water (O/W) emulsion is formed by emulsification of the astaxanthin solution with the aqueous phase containing the emulsifier. Then, pigment nanodispersion is achieved in a few cycles by means of high-pressure homogenizers. The lipophilic solvent is subsequently removed from the emulsion by rotary evaporation. Astaxanthin is crystallized in emulsion droplets during evaporation. Different surface-active biopolymers, such as polysaccharides and proteins, instead or with a combination of small molecular emulsifiers, such as lecithin, polysorbates, sugar esters and monoglycerides, can be used to stabilize various nanosized systems. The nature of the stabilizer significantly influences the stability of astaxanthin in prepared nanodispersion systems.

Tachaprutinun et al. [112] used poly(ethylene oxide)-4-methoxycinnamoylphthaloyl-chitosan (PCPLC) as a stabilizer. It yielded very good encapsulation efficiency (98%) at loading 40% (*w*/*w*). Moreover, the freeze-dried, astaxanthin-encapsulated PCPLC nanospheres showed good dispersibility in water, yielding stable aqueous suspensions of 300–320 nm nanoparticles. The thermal stability of astaxanthin in an aqueous environment was greatly improved upon PCPLC nanoencapsulation; i.e., the loss of olefinic functionality, observed when unencapsulated astaxanthin was heated at 70 °C for two hours, was prevented by PCPLC encapsulation. Ethyl cellulose (EC) and poly(vinylalcohol-co-vinyl-4-methoxycinnamate) (PB4) were also used as stabilizers. EC was totally inefficient, whereas PB4 exhibited poor encapsulation efficiency. The study showed that successful encapsulation required adequate compatibility between astaxanthin molecules and polymeric nanospheres.

Anarjan and Tan [113] prepared the astaxanthin nanodispersion with optimal physicochemical characteristics and the highest physicochemical stability. They used a three-component stabilizer system with 29% (*w*/*w*) polyoxyethylene sorbitan monolaurate, 6% (*w*/*w*) protein–sodium caseinate and 65% (*w*/*w*) polysaccharide–gum arabic. They achieved the maximum chemical stability of the astaxanthin nanodispersion by adding antioxidants, i.e., ascorbic acid and α-tocopherol.

Chemically stabilized astaxanthin nanodispersions can be incorporated into food products (such as beverages, soups, spreads), giving them the status of functional foods. The study by Mezquita et al. [114] suggested that astaxanthin oleoresin (AOE) could be successfully used to simulate apricot color in skimmed, semi-skimmed and whole milk. During seven-day storage in a domestic refrigerator at a temperature of 5 °C, there were no significant changes in three color coordinates L*, a* and b*. This indicated high stability of astaxanthin within the matrix. The water-dispersible emulsion from AOE was used to develop an orange-red color isotonic beverage (IB) with efficient antioxidant action [6]. It showed very good solubility in commercial colorless IBs, giving them an orange-red color, simulating that obtained with a synthetic pigment. However, in prepared PIBP (pigmented isotonic beverage prototype) samples stored at 30 °C and exposed to light, total degradation of astaxanthin was noted within 7 days. In contrast, in PIBP samples stored in the dark under refrigerated conditions (5 ± 2 °C), the pigment concentration decreased by 27% in relation to the initial value during the first week, and by the end of the storage time (91 days), it remained practically unchanged.

Tamjidi et al. [115] incorporated astaxanthin into nanostructured lipid carriers (NLC). NLC are O/W nanoemulsions in which major portions of the lipid phase are solid lipids. High encapsulation efficiency is achieved by effective immobilization of encapsulated lipophilic compounds. This improves their utilization, bioavailability and stability in fat-free and low-fat foods and transparent/opaque beverages. The researchers prepared optimal astaxanthin–NLC formulations consisting of 5wt.% lipid phase (5 mg lecithin as the emulsifier + 20 mg astaxanthin + 975 mg lipids (oleic acid as the liquid lipid and glyceryl behenate as the solid lipid)) and 95 wt.% aqueous phase (a solution of TWEEN 80 in phosphate buffer solution). Stability tests were carried out for astaxanthin-loaded NLCs (Ax-NLCs) in model beverages: Solutions of sucrose (pH 3.7), semi-actual (whey) and actual (non-alcoholic beer) for a period of 30–60 days storage at 6 or 20 °C were carried out. It was observed that the presence of sucrose improved the physical stability of Ax-NLCs in acidic model beverages. In whey, the average size of Ax-NLCs (94 nm) remained unchanged, and no astaxanthin loss was noted. However, carbonation and pasteurization processes of beer samples with NLCs added resulted in an increase in carrier particle size and turbidity as well as astaxanthin loss. Thus, NLCs should be added to CO_2_-free beverages after pasteurization.

Moreover, the stability tests carried out for CO_2_-free beers with Ax-NLCs added showed that storage of fortified beer at high temperatures and/or for a long time should be avoided. The desirability of organoleptic attributes of such beers decreased, but they were still acceptable. These results are significant in the context of applicability of nutraceutical-loaded NLCs in food and beverage systems.

## 5. Conclusions

Natural astaxanthin is a bioactive compound whose anti-oxidative activity and health-promoting properties, resulting from its unique structure, have been well-documented. However, due to its high price and limited sources, it is poorly known to food consumers and underestimated by food producers. For this reason, it is necessary to spread information about this pigment. The anti-oxidative potential of astaxanthin as well as its coloring properties enable food technologists to design a sensorily attractive assortment of functional foods and active packages. The incorporation of astaxanthin into nanodispersion systems is a promising alternative to the use of water-insoluble astaxanthin in water-based food systems.

*Haematoccocus pluvialis* freshwater algae, *Xanthophyllomyces dendrorhous* yeasts and crustacean byproducts are natural sources of astaxanthin. Although the cost of natural astaxanthin is high, researchers are conducting extensive investigations to reduce it due to the health-promoting and technological attractiveness of the pigment.

Synthetic astaxanthin is different from natural astaxanthin. It is a mixture of stereo-isomers. Some of them are not synthesized in nature, are less stable under technological conditions and have poor bioavailability. EU Regulation (EC) No. 1925/2006 on the addition of vitamins, minerals and other substances to foods does not permit the use of synthetic astaxanthin in food [64]. In the US, it does not have GRAS status, either. Its composition is different from the composition of natural astaxanthin. However, no research has indicated that synthetic astaxanthin might be harmful to humans or animals. It is commonly used in the feed industry, especially as a pink and red pigment in aquaculture.

## Figures and Tables

**Figure 1 molecules-26-02666-f001:**
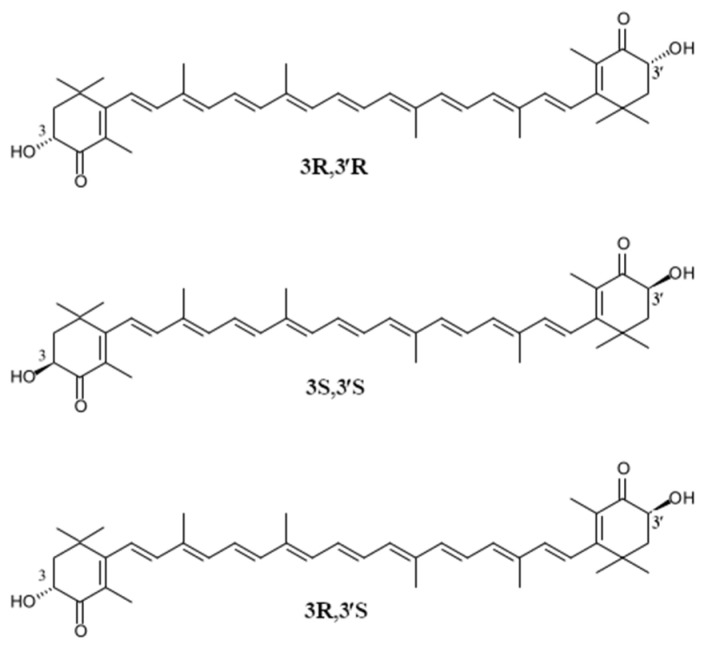
Configurational stereoisomers of astaxanthin.

**Figure 2 molecules-26-02666-f002:**
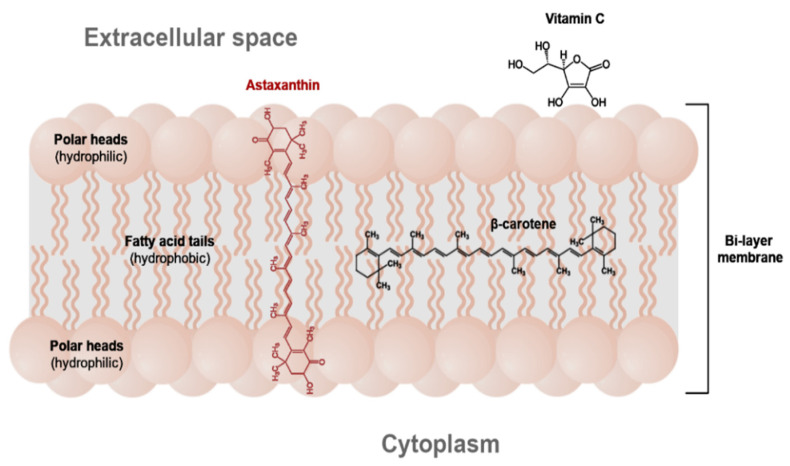
The location of astaxanthin and other antioxidants in the cell membrane (adapted from [15]).

**Figure 3 molecules-26-02666-f003:**
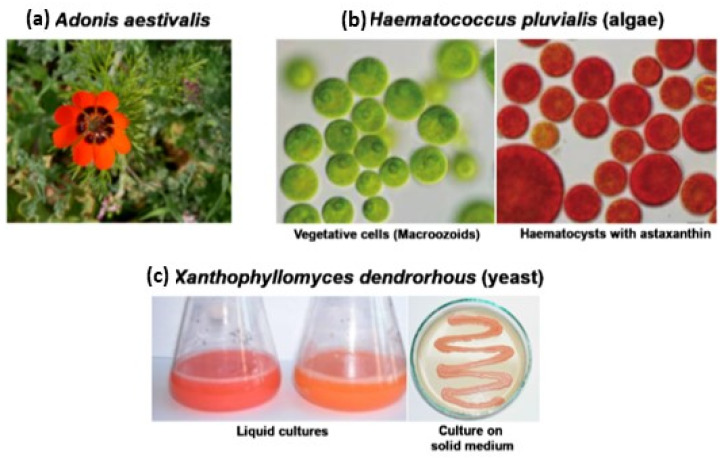
Natural sources of astaxanthin; (**a**) *Adonis* plants, (**b**) photo of *H. pluvialis alge,* (**c**) was adapted from [69].

**Figure 4 molecules-26-02666-f004:**
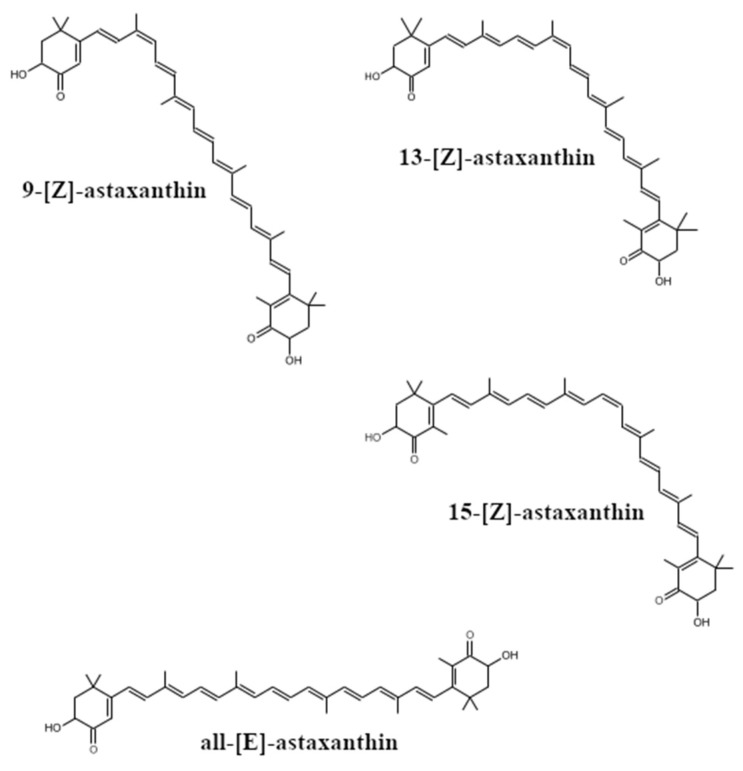
Astaxanthin E/Z isomers.

**Table 1 molecules-26-02666-t001:** Astaxanthin sources in nature and its configurational isomers.

Astaxanthin Source	Configurational Isomer [%]			References
	3S, 3′S	3R, 3′R	Meso Form	
*Xanthophyllomyces dendrorhous* (yeast)	-	100	-	[42]
*Hematococcus pluvialis* (algae)	100	-	-	[43]
Petels of *Adonis* spp.	100	-	-	[44,45]
Crustacyanine (lobster)	33	39	28	[39]
*Pandalus borealis* (shrimp)	12–25	-	50–53	[39]
Atlantic/Pacific salmon	78–85	12–17	2–6	[39]

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
