# Peer review of "Astaxanthin for the Food Industry"

_molecules, 2021, doi:10.3390/molecules26092666_

Round 1

Reviewer 1 Report

This is a thoughtful review of astaxanthin applications and production from natural sources and chemical synthesis, as well as the differences between products obtained in these ways. It will be useful for food and feed producents, cosmetic and pharmaceutical industries.

There are, however, minor problems with grammar, style and the choice of words in quite a few places:

Line 15. ‘health-promoting/benefiting” is better than “pro-healthy”- (also line 380).

Line 24. Use parentheses (…) for the chemical name instead of a dash before it.

Line 38. Comma before “thus”.

Line 44. It should be “benefits”.

Line 48. “structure”, not “composition”.

Line 58. Should it be “ecological water systems”?

Line 72. It should be “in DMSO”.

Line 82. Remove “inside the membrane”.

Line 92. “indicated/suggested” instead of “proved”, also in line 100” “Astaxanthin may prevent…”

Lines 105-109. The discrepancy between the recommended dose (4-12 mg/day, that is similar to other carotenoids) and the poor absorption of astaxanthin in humans (lines 377-382), compared to these carotenoids (lutein, lycopene, β-carotene) should be addressed somewhere in this review.

Line 109. “The” instead of “But”.

Line 130. Remove “simple”.

Table 1. It should be “Petals” of both Adonis species.

Line 139. “Studies showed that … [46] due to photodegradation.“

Line 142. It should be “bed-dried shrimp”.

Line 144-5. The sentence is too general. It should mention in what animals these positive effects were seen.

Line 151. “They may exhibit different activities from natural carotenoid isomers and cause…”

Line 152. What exactly the authors mean by “de novo synthesized (natural) astaxanthin”?

Line 157. It is “synthesis”, not “analysis”. It should be “obtaining astaxanthin”.

Line 178. It should be “DSM (Denmark), that developed CARYOPHYLL…., containing…”

Line 179. It should be “administered”.

Line 216. ”pigment obtained from microalgae.”

Line 231, 234. “pathway” is better than “route”.

Line 257. “source”, singular.

Line 282. “supercritical”, here and everywhere else.

Line 283. Remove colon after “as”.

Line 285. It should be “fermented and non-fermented liquid dairy products”, same on line 367.

Line 299. “while” instead of “related with”.

Line 302. It is “percent”.

Line 304. ”it is not advisable in aquaculture…”

Line 310 and everywhere else. “Byproducts”.

Line 315. “routinely used” (also line 459).

Line 325. Here, as well in many other places, the reference number is placed too far from the name of the authors, at the end of a long description. It should be “Pu et al [80]”

Line 336-7. Remove “(although … favourable)”.

Line 339-340. It should be “extraction from crustacean byproducts.”

Line 345-6. It should be “a promising separation technique for thermolabile compounds.” Also, join the next two sentences (“…and it is not necessary…”).

Line 363. Remove “wild and”. Is there synthetic astaxanthin in wild salmonids?

Line 369. Remove “incorporation”.

Line 389. Remove the sentence “Next, it needs to be fixed”.

Line 392. Comma before “polymer”. Line 393. It is “shelf life”.

Line 403. As above. Also ‘spray-drying’ here and in line 404, 410.

Line 405. “consists of”.

Line 412.  “acceptable” instead of “reasonably good”.

Line 424. “applications“ instead of “science”.

Line 433. “71-fold” instead of “times”.

Line 436. Remove “or “before ethanol, replace “and” with “or”.

Line 451. Remove “as an” before “agglomerate”.

Line 452. Remove “in the number size distribution”.

 Line 453. Comma after “ratio”.

Line 455. “related to”, not “with”.

Line 458. Comma after “stage”.

Line 461. “final product”.

Line 463. It should be “especially to 9-cis and 13-cis stereoisomers (Figure 4).” Skip the rest of the sentence.

Line 467. Remove “Researchers indicate that”.

Line 468. “and they oxidize…”

Line 469. “in” instead of “for”.

Line 472. “lipid peroxidation and ROS generation”.

Line 473. “than of all-trans astaxanthin”.

Line 475. Comma after DMSO.

Line 477. Comma after “safety”.

Line 484 “Astaxanthin” misspelled.

Line 485. Comma after “biopolymers” and “proteins”, as well as after “emulsifiers” in line 486.

Line 489. “psystems” – is it correct?

Line 495. Comma after “functionality and after “hours” in line 496.

Line 497. Remove space after “poly”.

Line 508. “introduced or incorporated” instead of “entered”.

Line 517. “However,” instead of ”But”.

Line 518 and 520. Incorrectly typed degree symbol - there should be no space after the number and the symbol should look like a superscript.

Line 522. There are two values, with no explanation, respectively.

Line 523. Remove “This is an important benefit for this product” and end this paragraph.

Line 524. Start a new paragraph “Tamjidi et al. [106] …’ and join it with the next two paragraphs, since they describe the same reference.

Line 525. Replace “is constituted by” with “are”.

Line 526. Remove dash after “effective”.

Line 528. Comma after “foods”.

 Lines 529-531. Use two different kind of brackets.

Line 533. It should be “Stability tests were carried out for astaxanthin…”

Line 534. “for a period”.

Line 535. Remove comma after “observed”, after “showed“ in line 542, after “decreased” in line 543.

Line 549. Comma after “sources”

Line 558. Remove “obtaining”.

Line 560-1. “stereo-isomers”, not “stereoi-somers”. What is “produced de novo” in this context?

Line 562.  “inassimilable” is awkward, “have poor bioavailability” sounds better.

The article should be carefully edited, because some faults are repeated, as indicated above, but not pointed out in every case by the reviewer. After corrections and addressing all concerns, the article should be ready for publication.

Author Response

Dear Sir or Madam
Thank you very much for your constructive and detailed review. I put the corrections in the attached file.

Reviewer 2 Report

The authors describe the possibly use of astaxanthin in food application in free and encapsulation form to increase its bioavalability.

Review is interesting

Line 72: introduce "dimethyl sulfoxide (DMSO")

Line 95: specify type 2 diabetes

From line 281 to 282: move the sentence where in the manuscript is described supercritical carbon dioxide extraction (SCCO2)

From line 338 to 349: describe better the SFE extraction technique and SCCO2

Line from 422 to 424: are  B-Cyclodextrin and its derivates food grade?

I suggest to explain better what are the cyclodextrins, the tipology used in food consulting the following paper:

Durante et al. 2020, Foods, DOI:10.3390/foods9111553

Durante et al. 2016, Food Chemistry, DOI:10.1016/j.foodchem.2015.12.073

Author Response

Dear Sir or Madam
Thank you very much for your constructive  review. I put the corrections in the attached file.

Reviewer 3 Report

The paper gives a good overview on the xanthophyll astaxanthin, presenting biological activities, commercial sources, industrial use as well as possibilities and limitations of its use in food technology. In addition, methods to stabilise astaxanthin by means of encapsulation techniques are presented, improving its bioavailability. Thus, the review will be a valuable source for scientists as well as for processors.
However, there are some points to be corrected to improve the quality of the manuscript:
Within the whole manuscript please check the writing of temperatures; sometimes they are given without blank between value and unit and otherwise with blank.
Page 1, line 20: one of the keywords is already mentioned in the title; please look for an alternative
Page 4, line 135: please change "source" into "sources"
Page 4, line 144: please change "function" into "functions"
Page 5, line 195: please insert a comma after "July 2020"
Page 6, line 219: please insert a comma after "China"
Page 7, line 249: please insert a comma after "At present"
Page 7, line 267: please insert a comma after "Then"
Page 7, line 274: please insert a comma after "In 2014"
Page 7, line 278: please insert a comma after "December 2017"
Page 7, line 282: please change "or a ethyl acetate" into "or ethyl acetate"
Page 7, line 284: please change three times the decimal comma into a decimal point and delete six times a blank prior to "%"
Page 9, line 401: please insert a comma after "Then"
Page 10, line 406: please insert a comma after "Then"
Page 10, line 411: please insert a comma after "emulsion"
Page 10, line 418: please change "Ciapara, et al" into "Ciapara et al."
Page 11, line 458: please insert a comma after "first stage"
Page 11, lines 463/471 and fig. 4: please use the more common (all-E)-/(Z)-nomenclature for astaxanthin isomers instead of the trans-/cis-nomenclature
Page 11, line 480: please insert a comma after "next stage"
Page 11, line 481: please insert a comma after "Then"
Page 11, line 484. please change "Astaxahnthin" into "Astaxanthin"
Page 12, line 489: please change "psystems" into "systems"
Page 12, line 514: please change "water -dispersible" into "water-dispersible" and change "used develop" into "used to develop"
Page 12, line 522: please give only one digit after decimal point for the two values
Page 12, line 533: please insert a comma after "Next"

Author Response

Dear Sir or Madam
Thank you very much for your detail and constructive  review. I put the corrections in the attached file.
